# How Do Floods and Drought Impact Economic Growth and Human Development at the Sub-National Level in India?

**Upali Amarasinghe \*, Giriraj Amarnath[ID], Niranga Alahacoon and Surajit Ghosh**

International Water Management Institute (IWMI), 127 Sunil Mawatha, Pelawatte, Colombo 10120, Sri Lanka;
a.giriraj@cgiar.org (G.A.); n.alahacoon@cgiar.org (N.A.); s.ghosh@cgiar.org (S.G.)
\* Correspondence: u.amarasinghe@cgiar.org; Tel.: +94-11-288-0000

**Abstract:** This paper tries to shift the focus of research on the impact of natural disasters on economic growth from global and national levels to sub-national levels. Inadequate sub-national level information is a significant lacuna for planning spatially targeted climate change adaptation investments. A fixed-effect panel regression analyses of 19 states from 2001 to 2015 assess the impacts of exposure to floods and droughts on the growth of gross state domestic product (GSDP) and human development index (HDI) in India. The flood and drought exposure are estimated using satellite data. The 19 states comprise 95% of the population and contribute 93% to the national GDP. The results show that floods indeed expose a large area, but droughts have the most significant impacts at the sub-national level. The most affected GSDPs are in the non-agriculture sectors, positively by the floods and negatively by droughts. No significant influence on human development may be due to substantial investment on mitigation of flood and drought impacts and their influence on better income, health, and education conditions. Because some Indian states still have a large geographical area, profiling disasters impacts at even smaller sub-national units such as districts can lead to effective targeted mitigation and adaptation activities, reduce shocks, and accelerate income growth and human development.

**Keywords:** floods; droughts; gross domestic product; HDI; satellite data; states; India

## 1. Introduction

Natural hazards cause severe economic losses worldwide [1]. Over the last 20 years, the global financial damages due to natural hazards are over 2,440 billion USD (EM-DAT database https://www.emdat.be/). About 74% of these natural hazards were related to water. Floods and storms accounted for 69% of the financial damages, exposing over three billion people and causing 166,000 deaths globally. Between 2000 and 2020, South Asia alone experienced 11% of the world's natural disasters and 12% of floods and droughts, exposing over 700 million people and 190 million ha of agricultural land [2,3].

Water-related disasters from floods and droughts can constrain economic development in many countries [4–8]. However, some studies with national-level data show that floods and droughts have no discernible impacts on long-term economic growth [9,10]. Yet, some other studies show considerable variations of the impact on income growth [11,12]. Regardless of these contrasting findings, it is clear that the severity and frequency of floods and droughts are increasing at the sub-national level [13–15]. Without adequate information on impacts at the sub-national level, it is difficult to develop proper adaptation and mitigation plans. In the absence of adaptation plans, disasters with increasing intensity can affect millions of more people and inhibit regional economic growth and human development [16].

The sub-national variability of natural disasters is especially prevalent in large countries. India, the second-most populous country in the world, has 1.4 billion people spread across 28 major states

and eight small union territories in an area of 32,87,263 km$^2$. These states have considerable spatial and temporal variation in climate. Floods and droughts accounted for 51% of all natural hazards and 76% of the damages caused in India between 2000 and 2020 [3]. Other natural hazards—earthquakes, landslides, wildfires, storms, and high temperature and rainfall events—affect several regions [2]. However, a lack of reliable information on the impacts of natural disasters on economic growth and human development at the sub-national level is a significant lacuna for regional policy planning and targeted climate investments [14,17].

Floods and droughts are a recurrent phenomenon in India [18]. The southwest monsoon rains, between June and October, account for over 70% of annual precipitation in states located in many large river basins [19]. The large river basins, such as the Indus, Ganges, and Brahmaputra, generate significant monsoon runoff leading to massive flooding in the plains. About 43% of the Indian population is prone to recurrent floods [2]. The northeast monsoon period (November to March) has mainly dry weather, and a year with below-average monsoon causes droughts in many regions. Annually, droughts also expose a similar percentage of the population [2]. Those hit hardest are the rural areas and with agriculture-dependent livelihoods, especially where rainfall is the only source of water supply for agricultural production. These areas include agriculture in rain-fed, groundwater, and minor tank irrigation areas consisting of about two-thirds of the total cropped area [20].

In 2015, agriculture contributed to 16% of the GDP in India, varying from 6% to 35% across states [19]. However, 30% of the population has agriculture-dependent livelihoods, which are vulnerable to recurrent floods and droughts. The share from the agriculture sector to the overall GDP is decreasing gradually and was less than 15% in many states in 2018 [19]. Therefore, it is essential to assess the impacts of natural disasters on the growth of non-agriculture sectors to assess the implications on overall economic growth [9].

Physical water scarcity is already prevalent in many regions in India [21], and in these locations, the existing water resources are inadequate to catalyze further water resources infrastructure development. The severity of water scarcities will increase in the future. In general, recurrent water scarcities can impact economic growth and human development [22], and natural hazards with increased severity can exacerbate this situation.

This paper attempts to shift the focus from the national to the sub-national level assessments on the relationship between natural disasters and economic growth. Specifically, it assesses how floods and droughts affect the gross state domestic product (GSDP) and human development in India. The analysis for the first time uses a gridded time-series information of floods and droughts exposed areas derived from fine-scale satellite data [2]. Most national studies use damages and losses in the EM-DAT dataset, which does not show the spatial spread of floods and droughts. The study considers 19 major states, which account for 95% of the total population and 93% of the national GDP in 2015. Excluded from the analysis are union territories and northeastern states, where reliable data are not available.

Sub-national analysis of impacts of floods and droughts exposure is vital given the massive investments on mitigation and adaptation to natural hazards. Since 2007, the investments in flood protection in different river basins and states in India are over one billion USD (http://cwc.gov.in/fm-projects). These include large river basins such as Indus, Ganges, Brahmaputra, Mahanadi, and Godavari, which have chronic flood-related issues. These investments envisage reducing the impacts in flood-prone riparian states such as Punjab and Haryana in the north-western region, Uttar Pradesh in the northern region, and Bihar, Odisha, and Assam in the eastern and north-eastern regions. However, the water stress in the dry season is also frequent in these riparian states. Alam et al. [23] showed that the spatial spread of floods can be a good source for groundwater recharge, which augment water supply for the water stress periods enhancing the resilience of both the agricultural and industrial sectors. Moreover, the flood exposed areas and their impacts are useful for designing bundled solutions of index insurance with climate information and seed systems as agricultural risk management tools [19], which are being pilot tested in Eastern India. The impacts

of drought exposure on different economic sectors are useful information for crop and agricultural diversification [24], and planning interventions for drought proofing and enhancing water security by both the public and private sector [25–27].

In the next section, we present the data and methodology used for the study, followed by an analysis of flood and drought trends in the exposed areas and consequent damages. The fourth section is about the impacts of floods and droughts on the per-capita GSDP and human development index (HDI). We conclude the paper with a discussion on the results and policy implications.

## 2. Materials and Methods

### 2.1. Literature Review

A review of recent literature assessing the impacts of natural disasters on economic growth shows a contrasting picture. Botzen et al. [28] review show that most natural disasters have direct economic consequences, but only the less diversified economies have severe indirect effects leading to slowing economic growth. Shabnam [11], however, shows a contrasting picture. A fixed effect panel data analysis of 197 countries from 1960 to 2010 shows an adverse effect of the number of people affected by floods on long-term economic growth.

Brown et al. [5], using a fixed effect regressions of a panel of 133 countries from 1961 to 2003, showed that droughts and floods constraint GDP growth. The findings of this report show that a one percent increase in flood and drought-affected areas reduce economic growth by 2.7% and 1.8%, respectively. Brown et al. study confirmed the findings of Grey and Sadoff [6] on enhancing water security for economic growth.

Cavello et al. [9] also used panel data from 1970 to 2008 of 198 countries but showed only extremely large disasters impact long-term economic growth. However, when controlled for political upheavals, even large disasters have no impact on economic growth. The source of data for their study is EMT-Data set, and it is possible that damage and loss indicators in these data are correlated with GDP per capita, and do not capture the spatial spread within countries. Felbermayr and Jasmin [29], using geophysical and meteorological data, showed that natural disaster affects economic growth, and only geophysical disasters mostly affect the growth in developing countries. They also used fixed effect panel regression with data from 108 countries from 1979 to 2008.

Loayza et al. [12], using the generalized method of moments (GMM) on a cross country panel from 1990 to 2010, showed varying impacts of disasters on the economic growth of different sectors. Panwar and Sen [23], analyzing a panel data of 102 countries from 1981 to 2015 with a system of generalized method of moments approach, also showed similar results of natural disasters on economic growth. This analysis used types and frequency of flood, droughts, storms, and earthquakes to characterize the occurrence of disasters.

Most impact studies on disasters and economic growth used aggregated national-level statistics of damages and losses using the information in the EMT-Data. However, there are only a few studies in the literature assessing natural disasters' impacts at sub-national economic growth. A recent study by Tang et al. [30] used panel regression to assess the impacts of natural disasters in three regions. Using lagged terms of all control variables to control endogeneity, the study shows the heterogeneity of regional impacts, with meteorological disasters mostly affecting the central and eastern regions. Panwar and Sen [31], using an augmented panel vector auto-regression model across 24 states of India, shows that floods have a negative effect on the short-term growth of all economic sectors except the agriculture. Only the severe floods have a statistically negative or non-significant effect.

### 2.2. Data

The 19 states in the study cover most areas of large river basins in India (Figure 1, Table 1). The only exception is the northeastern states, excluding Assam, which comprise a substantial area of the Brahmaputra river basin. With more than 600 Bm$^3$ of annual runoff, floods are a recurrent

phenomenon in the Brahmaputra. The state of Assam, located in the downstream of the Brahmaputra and the most affected by its floods, is included in this study. The study first assesses the trends of exposed areas and GSDP to floods and droughts. Next, it conducts a panel data analysis at the state level to evaluate the impacts of floods and droughts on the GSDP, state HDI, and economic growth.

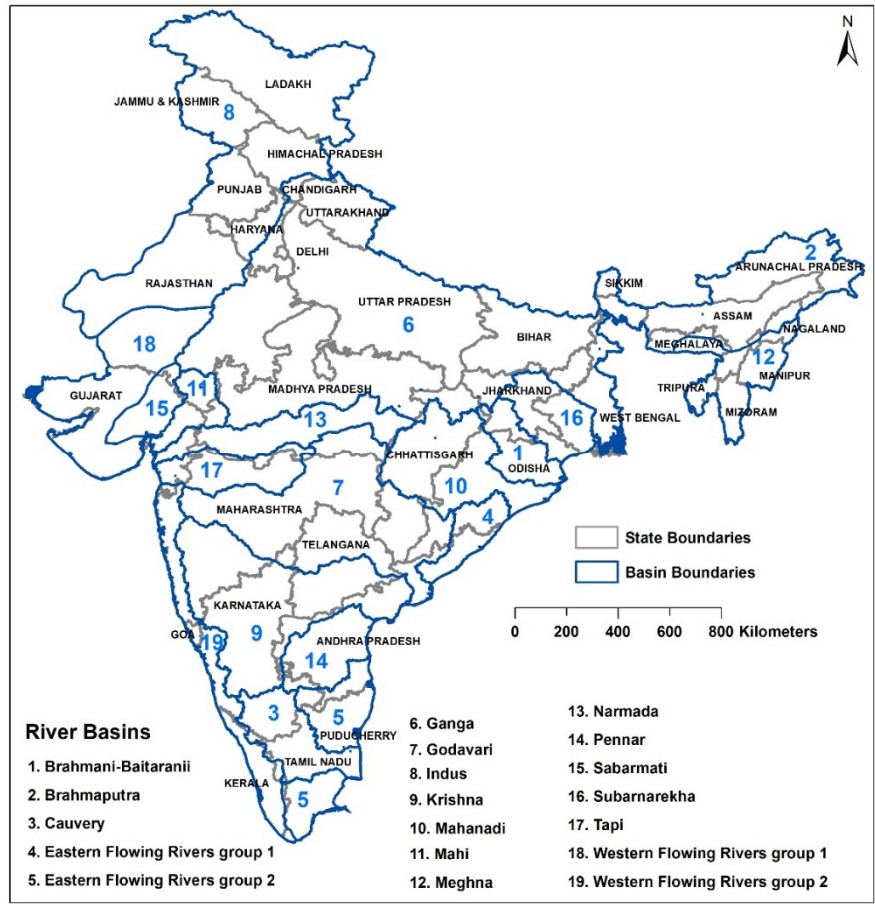

**Figure 1.** State and river basins of India.

The selected states fall into four different groups based on geography, hydrogeology, and GSDP; namely, the states in the eastern region (ER), southern and western region (SWR), northern and central region (NCR), and northwestern (NWR) regions.

The states in ER and NCR have low per-capita GSDP, but being located in the downstream of large river basins, the dynamics of floods in the ER states are quite different from those in the NCR states. Therefore, the study considers the ER states separately from the NCR group. In comparison, the SWR and NWR states have higher per-capita GSDP. Hydro-geologically, the NWR states have alluvial aquifers, and the SWR states mainly have hard-rock aquifers. These differences affect water availability and use patterns in the agriculture sector. Therefore, the study considers NWR states in a separate group.

The hazards related data for the analyses are the gridded information of floods and droughts-exposed areas (Figure 2) estimated from satellite data [2]. This information is the first of its kind multi-year high-resolution gridded estimates available on floods and droughts-exposed areas and GSDP. Data used for generating exposure data are given in Table 2.

**Table 1.** Key statistics of the 19 states on the population, gross state domestic product (GSDP) and human development index (HDI).

| Region | State | Population (Millions) | | GSDP [1]/pc (USD) | | Agriculture GSDP [1] (USD) | | HDI | |
|---|---|---|---|---|---|---|---|---|---|
| | | 2013–2015 | Annual Growth [2] since 2002 | 2013–2015 | Annual Growth [2] since 2002 | % of Total GSDP in 2013–2015 | Annual Growth [2] since 2002 | 2001–2003 | 2013–2015 |
| Eastern Region (ER) | Assam | 27.1 | 1.4 | 3401 | 3.6 | 19 | 0.2 | 0.490 | 0.594 |
| | Bihar | 84.9 | 1.6 | 2356 | 5.2 | 22 | 1.5 | 0.405 | 0.491 |
| | Odisha | 37.3 | 1.0 | 5076 | 7.8 | 16 | 3.7 | 0.400 | 0.485 |
| | West Bengal | 81.2 | 1.1 | 5360 | 5.2 | 19 | 2.9 | 0.540 | 0.655 |
| Northern and Central Region (NCR) | Chhattisgarh | 21.3 | 1.5 | 3461 | 5.0 | 16 | 3.5 | 0.507 | 0.615 |
| | Jharkhand | 27.5 | 1.6 | 2166 | 5.3 | 15 | 3.8 | 0.416 | 0.504 |
| | Madhya Pradesh | 61.5 | 1.7 | 3752 | 4.8 | 31 | 6.1 | 0.415 | 0.503 |
| | Uttar Pradesh | 169.3 | 1.8 | 3357 | 4.7 | 26 | 2.6 | 0.419 | 0.509 |
| | Uttarakhand | 8.6 | 1.6 | 3331 | 4.9 | 11 | −1.6 | 0.538 | 0.653 |
| Southern and Western Region (SWR) | Andhra Pradesh | 77.0 | 1.0 | 7152 | 7.3 | 21 | 5.2 | 0.522 | 0.633 |
| | Karnataka | 53.6 | 1.2 | 6652 | 6.9 | 14 | 2.1 | 0.573 | 0.695 |
| | Kerala | 32.0 | 0.7 | 8693 | 7.3 | 9 | 2.5 | 0.869 | 0.998 |
| | Tamil Nadu | 63.3 | 0.8 | 7607 | 7.0 | 10 | 4.1 | 0.628 | 0.762 |
| | Gujarat | 51.6 | 1.5 | 8097 | 6.6 | 15 | 6.3 | 0.580 | 0.704 |
| | Maharashtra | 98.3 | 1.5 | 8776 | 6.5 | 8 | 1.7 | 0.630 | 0.764 |
| | Rajasthan | 57.6 | 1.8 | 4363 | 5.6 | 25 | 4.8 | 0.479 | 0.581 |
| Northwest Region (NWR) | Haryana | 21.5 | 1.8 | 10066 | 6.3 | 19 | 2.8 | 0.608 | 0.737 |
| | Himachal Pradesh | 6.2 | 1.1 | 8404 | 5.6 | 12 | 0.8 | 0.718 | 0.871 |
| | Punjab | 24.7 | 1.2 | 7956 | 4.9 | 26 | 2.2 | 0.664 | 0.806 |

Notes: [1]—GSDP is PPP (purchasing power parity) in constant 2010 USD; [2] —growth is the average annual growth.

**Table 2.** List of data sources used in the impact analysis.

| Hazard | Dataset | Period | Spatial/Temporal Resolution | Source |
|---|---|---|---|---|
| Floods | Moderate Resolution Imaging Spectroradiometer (MODIS) surface reflectance product (MOD09A1) | 2001–2013 | 500 m/8 days | National Aeronautics and Space Administration (NASA) [1]; surface reflectance product (Amarnath et al. 2012); |
| Droughts | MODIS surface reflectance product (MOD09A1) | 2001–2013 | 500 m/8 days | NASA [2]; Amarnath 2014b |
| Gross state domestic product | GSDP exposed to floods and droughts | 2001–2013 | State/annual | |
| HDI | Human Development Index (HDI) | | State/Annual | |

Sources: [1] https://lpdaac.usgs.gov/dataset_discovery/modis/modis_products_table/mod09a1. [2] http://sedac.ciesin.columbia.edu/data/collection/gpw-v3.

The flood-exposed area (FLEA) is the maximum area exposed to floods during a year (Figure 2a), which is a good indicator of the extent of inundation. The FLEA of a state is the total area of flood-exposed pixels. However, some areas, such as those located in the ER in the lower reaches of the Ganges and Brahmaputra, can have several floods during the monsoon season. These waves of floods may cause more damages than a single flood event. Yet, information neither on the intensity nor on the duration of multiple floods was available for this analysis.

Estimates of the areas exposed to floods and droughts are at the pixel level of 500 m spatial resolution [2]. It used 8-day composite images for the period from 2001-2015 from the MODIS Surface Reflectance product (MOD09A1) [32,33] from NASA's Earth Observing System Data and Information System (EOSDIS). The flood-exposed pixels are identified using the flood inundation mapping algorithms [34]. The algorithms, with the aid of digital elevation models (DEMs), enable us to classify pixels into temporary flooded or permanent water bodies. The FLEA of a state is the total area of temporarily flooded pixels of 8-day MODIS images between 2001 and 2013.

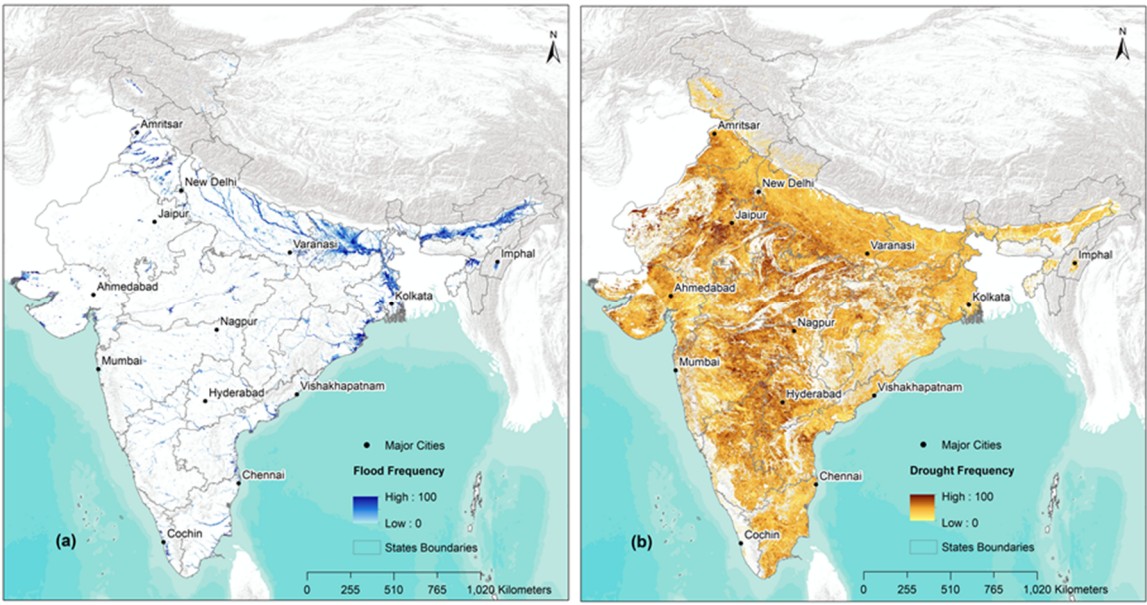

**Figure 2.** Maximum exposed area to (**a**) floods and (**b**) droughts using combined hazard and population gridded data.

The drought-exposed area (DREA) is the maximum area exposed to meteorological or agricultural droughts (Figure 2b). These droughts, during critical crop growth periods, can reduce agricultural production significantly. Water scarcities associated with droughts affect non-agricultural activities. The Normalized Difference Drought Index (NDDI) identifies the drought-exposed pixels. The NDDI is a function of the Normalized Difference Vegetation Index (NDVI), and Normalized Difference Water Index (NDWI) estimated from satellite images. NDVI and NWDI determine the level of vegetation and the water content in leaves or water bodies, respectively.

The analysis uses these gridded estimates at the pixel level to aggregate areas and GSDP's exposed to floods, and droughts at the state level. The HDI is available at the state level only.

## 2.3. Econometric Analysis

This assessment starts with an explorative data analysis of the trends of area and GSDP exposure to floods and droughts in different regions. Monsoon rain is the main factor that determines floods (mainly between June and September) and agricultural droughts thereafter (October to May). For floods, the analysis considers data of 2003, 2010 and 2013, years with above average monsoon rainfall. For droughts, the focus is on 2002, 2009, and 2015, the years with below average monsoon rainfall.

Next, using state and time fixed effect panel regression model (Equations (1) and (2)), the analysis assesses the extent to which flood and drought exposed areas influence the GSDP exposed to floods and droughts. In some regions, floods and droughts can have carry-over effects on the economic growth. We test the carry-over effect using the previous year's (lag 1) flood and droughts exposed areas as explanatory variables. It is likely that there is additional time variant factors that affect the GSDP; to capture that we include a time fixed effect.

$$FLEGDP_{it} = \alpha_1 FLEA_{it} + \alpha_2 FLEA_{it-1} + \alpha_3 DREA_{it} + \alpha_4 + \gamma_i + \delta_t + \epsilon_{it} \ldots \ldots \quad (1)$$

$$DREGDP_{it} = \beta_1 DREA_{it} + \beta_2 FLEA_{it-1} + \beta_3 DREA_{it} + \beta_4 DREA_{it-1} + \gamma_i + \delta_t + \epsilon_{it} \ldots \ldots \quad (2)$$

where $FLEGDP_{it}$ and $DREGDP_{it}$ are the flood and drought exposed GDP in the $i$th state at the $t$th period. The $FLEA_{it}$ and $DREA_{it}$ are flood and drought exposed areas, $X_{it}$ s are other exogenous variables, which includes only the time (year) in Equations (1) and (2). The $\alpha$'s and $\beta$'s are regression coefficients, and $\gamma$'s and $\delta$ are state and time variant variables. The subscripts $i$ and $t$ vary across states and time: $i = 1..,19$ and $t = 2001,..,2015$.

Finally, the analysis assesses the impacts of current and previous years flood and drought exposed areas on sectoral (agricultural and non-agricultural) and total GSDP per person and HDI. Here also we use state and time fixed effect panel regression models. The assessment considers the following regression models

$$\left.\begin{array}{c} lnAGDPPC_{it} \\ lnNAGDPPC_{it} \\ lnTGDPPC_{it} \\ lnHDI_{it} \end{array}\right\} = \alpha_1 FLEA_{it} + \alpha_2 DREA_{it} + \alpha_3 FLEA_{it-1} + \alpha_3 DREA_{it} \\ + \alpha_4 DREA_{it-1} + \gamma_i + \delta_t + \epsilon_{it} \ldots \ldots \quad (3)$$

where AGGDPPC, NAGDPPC, and GDPPC are agriculture, non-agriculture, and total GDP per capita; HDI is the human development indicator, $X_{it}$'s are other exogenous variables. The $\alpha$'s are regression coefficients, and $\gamma$'s are dummy indicators for the state variables. The subscripts $i$ and $t$ vary across states and time: $i = 1..,19$ and $t = 2001,..,2015$.

In the regression models in equations 1 to 4, the floods and droughts exposed areas (FLEAs and DREAs) are the main predictor variables. In general, the growth of GSDP and human development enhances the adaptation and mitigation capacity to floods and droughts, which in turn affect the FLEAs and DREAs. Thus, the error terms in regressions are likely to correlate with the floods and

droughts related predictors. Therefore, we use two-stage least-squares regression analysis with monthly rainfall and deviation of monsoon and non-monsoon season rainfall from the averages as instrumental variables. We test the endogeniety of predictors with Wu–Hausman test, and suitability of and overidentifibility of the models with instrument variables with Wald and Sargens test respectively.

## 3. Results

### 3.1. Trends of Floods and Droughts Exposed Area and GSDP

Floods affected the ER states the most and had substantial variation across other states. Figure 3A shows the FLEA and GDP in 3 years (2003, 2010, 2013) that received above-average monsoon and annual rainfall. The Y-axis indicates the FLEA as a percentage of the total area of the state, and the size of the bubble indicates the GSDP of the exposed area to floods as a percentage of the total GSDP. It shows the following:

The states of AS (Assam), BI (Bihar), OR (Odisha), and WB (West Bengal) are in the eastern region; Andhra Pradesh (AP), Tamil Nadu (TN), Karnataka (KT) and Kerala (KE) are in the southern region; MH (Maharashtra), GJ (Gujarat), and Rajasthan (RJ) are in the western region; JH (Jharkhand), CH (Chhattisgarh), MP (Madhya Pradesh) are in the central region, and PU (Punjab) and HY (Haryana) are in the north-western region.

- Bihar and Assam in the eastern region (ER) have some of the highest exposure to floods. They also have some of the lowest levels of GSDP. However, their levels of exposure to floods have more than halved in the last two decades.
- In general, ER states, except Odisha, still have substantially high exposure to floods. Odisha has relatively lower flood exposure (about 4%). However, Odisha, along with Andhra Pradesh and Tamil Nadu, have high exposure to cyclones [2]. Cyclones related flash flooding inundates the densely populated urban centers and contributes to higher losses in the industrial and service sectors.
- The NCR states, such as Madhya Pradesh, Jharkhand and Uttar Pradesh have some of the lowest per capita GSDP but have relatively little exposure to floods.
- Punjab and Haryana in the NWR and Gujarat in the SWR have different trends of flood exposure, where area and GSDP exposed to floods after 2010 are increasing.

In India, the DREA and GSDP were substantially higher than those exposed to floods. Figure 3B shows the exposure to droughts of three years with below-average rainfall. The exposure to droughts was most extensive in the southern, western, and central states. However, the extent decreased over time in most states.

- Gujarat and Rajasthan in the WR had the most extensive exposure to droughts in the early 2000s; more than 35% of the area and 45% of the GSDP in 2002, i.e., 45% GSDP was generated in the area exposed to droughts. However, the area exposed to droughts in these states decreased substantially to less than 5% of the total area in 2015.
- The other high-income (GSDP/capita) states such as Tamil Nadu, Andhra Pradesh, Karnataka, and Maharashtra too, have substantially lower exposure to droughts in 2015.
- Punjab and Haryana states in the NWR too showed a similar drought-exposure pattern. However, although these states, along with Gujarat, may have mitigated drought impacts with enhanced infrastructure, the exposures to floods perhaps have increased over time (Figure 3A).
- The NCR states of Madhya Pradesh and Jharkhand have also reduced exposure to droughts substantially, although they had a low per-capita GSDP initially.
- The DREA and GSDP in the NCR and ER states are relatively lower. However, the volume of GSDP exposed to drought has increased between 2002 to 2015. These states may have focused more on mitigating exposure to floods than droughts or addressing other pressing social issues. These states have some of the lowest per-capita GSDP and the highest poverty rates in the country.

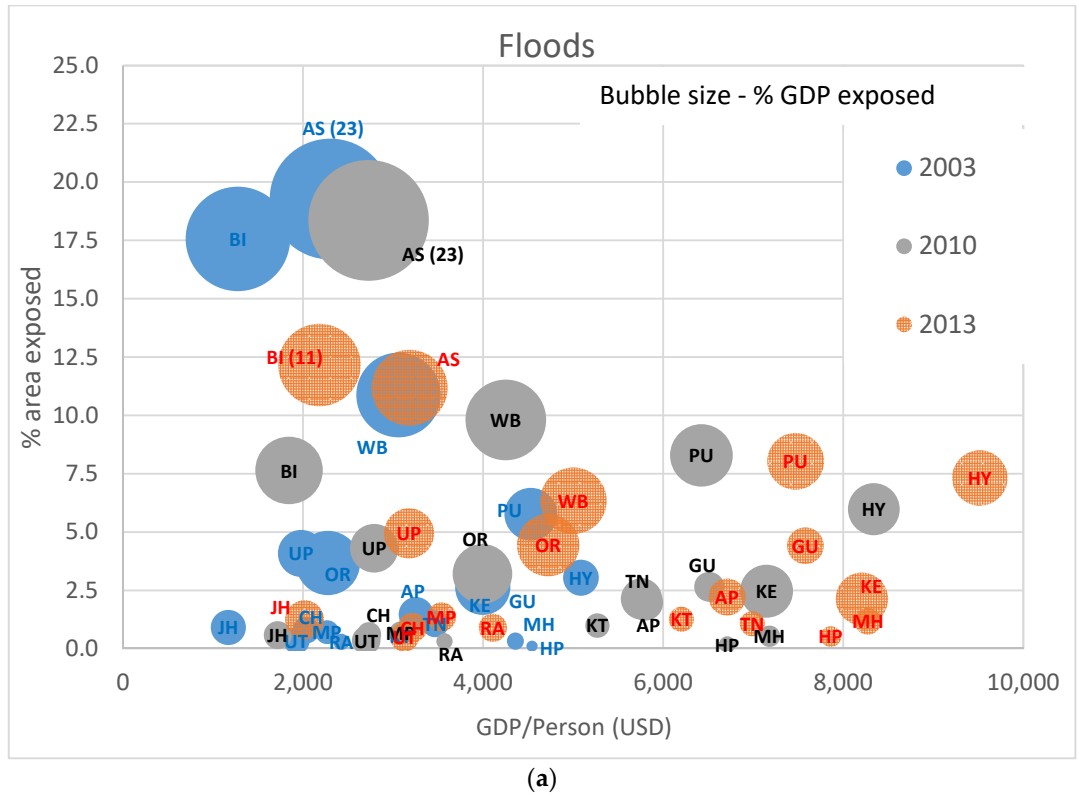

(a)

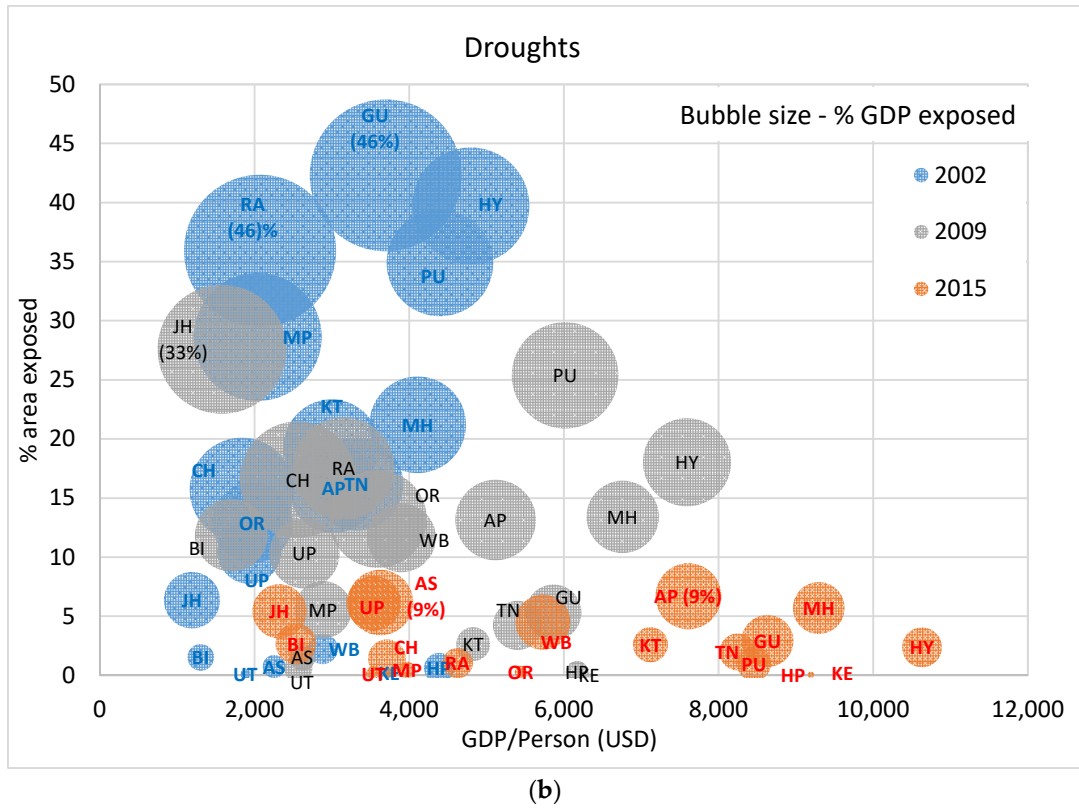

(b)

**Figure 3.** (**a**) Percent of area and GDP exposed to floods. (**b**) Percent of area and GDP exposed to droughts.

The regression analysis also shows that areas exposed to floods and droughts have substantial GSDP exposure too (Tables 3 and 4). The Wu–Hausman test shows that the flood exposed area has an endogeneity issue. However, the Wald and Sargen tests shows that the instrumental variables we used for modelling the endogenity are very weak, and become overidentified if used in the estimation. Therfore, in the absence of any other instrument variables, we present only the results of time and state fixed panel regressions.

**Table 3.** Statistical significance of the relationship of flood exposed areas and GSDP.

| Explanatory Variables | Dependent Variable—Percent GSDP Exposed to Floods | | | | |
|:---:|:---:|:---:|:---:|:---:|:---:|
| | All States | ER | SWR | NCR | NWR |
| % area exposed to floods | 1.140 *** | 1.160 ** | −0.847 ** | 0.777 ** | 0.679 |
| % area exposed to droughts | −0.001 | −0.021 | −0.006 | 0.002 | −0.007 |
| Lag 1 (% area exposed to floods) | −0.047 | −0.082 | −0.319 ** | −0.028 | 0.007 |
| Lag1 (% area exposed to droughts) | −0.001 | −0.068 | 0.001 | −0.007 | 0.005 |
| Adjusted $R^2$ | 0.95 | 0.92 | 0.98 | 0.96 | 0.97 |

*Notes: \*, \*\*, and \*\*\* indicate statistical significance at 0.1, 0.05, and 0.005 probability level.*

**Table 4.** Statistical significance of the relationship of drought exposed areas and GSDP.

| Explanatory Variables | Dependent Variable—Percent GSDP Exposed to Droughts | | | | |
|:---:|:---:|:---:|:---:|:---:|:---:|
| | All States | ER | SWR | NCR | NWR |
| % area exposed to floods | −0.128 | −0.010 | 0.250 | 0.0756 | 0.099 |
| % area exposed to droughts | 1.02 ** | 1.060 * | 1.120 ** | 1.100 ** | 0.743 *** |
| Lag (% area exposed to floods) | 0.004 | −0.183 ** | 0.065 | −0.38 | 0.483 ** |
| Lag (% area exposed to droughts) | 0.004 | 0.048 | −0.01 | 0.063 | −0.007 |
| Adjusted $R^2$ | 0.95 | 0.98 | 0.98 | 0.96 | 0.97 |

*Notes: \*, \*\*, and \*\*\* indicate statistical significance at 0.1, 0.05, and 0.005 probability level.*

On average, a 1% increase in the FLEA increased flood exposed GSDP by 1.14%, meaning that FLEAs mainly had higher GSDP. This fact was the actual situation in most river basins where most industrial and service facilities were near-surface water resources such as rivers and tributaries. Similarly, a 1% increase in DREA increased the drought-exposed GSDP by 1.02%. This estimate indicates that increasing the exposure of areas with high GSDP to droughts even when the total exposed area decreased between 2000 and 2020. The increasing GSDP in all states, albeit gradual, contributed to this anomaly.

In addition, the following was seen:

- FLEAs is generally associated with significantly higher GSDP, except in the NWR. The dichotomy in NWR could be due to increased flood exposure of areas with low non-agriculture activities.
- There seemed to be statistically significant—abrupt or gradual—impacts of flood mitigation investments, especially in the ER, and SWR (indicated by the negative and positively significant coefficients of lag one flood-affected area). In the ER, this could be due to damages caused by the magnitude of and damages casued by the floods. In the NWR, higher flood spread contributes to natural groundwater recharge, and that contributed to higher GDP in the exposed areas.
- A spread of floods in drought-stricken areas contributed to groundwater recharge, which was the primary source of water for most economic activities. However, the hypothesis tested with lag one DREA showed no statistically significant relationship. The aggregated state-level data used in the analysis generally hid these locally concentrated positive impacts. In 2005, groundwater contributed to 60% of the total water withdrawals and footprints (i.e., evapotranspiration) in India [35]. However, this footprint could be significantly higher now with either decreasing or stagnating trends of surface irrigation in many states.

Another notable observation was the decreased flood-exposed and increased drought-exposed GSDP over time in many states. These may be due to the effects of flood and drought mitigation and adaptation interventions and the increasing GSDP, regardless of floods and droughts, mainly contributed by fast-growing non-agricultural economic activities by the industrial and service sectors.

### 3.2. Impacts of Floods and Droughts

### 3.2.1. Sectoral GSDPs

Regression analysis confirms that the impacts of floods and droughts on the GSDP varied across economic sectors and regions. The results of the fixed effect panel regression analysis at the state level for per capita agriculture, non-agriculture, and total GSDP are given in Tables 5–7. The regression analysis considered the NCR, and SWR as single regions, as they had similar levels of per-capita GSDP. The analysis showed the following:

- The FLEA generally had a positive but no statistically significant impact on the GDP of different sectors in different regions. However, there was a significant favorable influence of flood exposure for non-agriculture sector GSDP in the NWR. Floods that occurred with the monsoon period usually enhanced groundwater recharge, which increased water availability in the post-monsoon period for the NWR. At present, groundwater contributes to about two-thirds of the water withdrawals for the agriculture and non-agriculture sectors.

- These results are consistent with the findings of [12], which assessed impacts separately on sectoral GDPs. It reported significant negative impacts of exposure of droughts and significant positive effects of exposure to floods on the economic growth of different sectors and the overall per-capita GDP in developing countries. Brown et al. (2013), [5] using an analysis of cross-country data, reported that a 1% increase in exposure to drought reduces GDP growth by 2.7%. However, Brown et al.'s findings on the impacts of floods are precisely the opposite; it reported that a 1% increased exposure to floods reduced GDP by 1.8%.

- The estimated average annual losses due to floods as a percentage of total GDP over the last decade (2005–2015) was only 0.4% (based on EMDT data). Floods can, indeed, cause damage to infrastructure. However, the impacts of the vast spread of floods on groundwater recharge outweighed the costs caused by it in some regions, because the water supply, especially groundwater, in 8 months of the non-rainy period was critical to livelihood and economic activities in many states.

**Table 5.** Regressions on flood and drought-affected areas with agriculture per-capita GSDP.

| Explanatory Variables | Dependent Variable—ln (Agriculture Per−Capita GSDP) | | | | |
|---|---|---|---|---|---|
| | All States | ER | SWR | NCR | NWR |
| % area exposed to floods | 0.003 | 0.002 | 0.003 | −0.001 | 0.116 *** |
| % area exposed to droughts | 0.004 *** | −0.005 * | −0.002 | −0.003 | 0.001 |
| Lag (% area exposed to floods) | 0.003 | −0.006 | 0.026 | 0.058 | 0.036 ** |
| Lag (% area exposed to droughts) | 0.004 *** | −0.004 * | −0.002 | −0.001 | 0.007 |
| Adjusted R$^2$ | 0.94 | 0.93 | 0.87 | 0.97 | 0.96 |

*Notes:* ***, **, and * indicate statistical significance at 0.005, 0.05, and 0.1 probability levels.

**Table 6.** Regressions on flood and drought-affected areas with non-agriculture per-capita GSDP.

| Explanatory Variables | Dependent Variable—ln (non-Agriculture per-Capita GSDP) | | | | |
|---|---|---|---|---|---|
| | All States | ER | SWR | NCR | NWR |
| % area exposed to floods | 0.002 | −0.002 | 0.021 * | 0.015 | 0.005 |
| % area exposed to droughts | −0.001 ** | −0.005 ** | −0.001 | 0.003 | −0.024 |
| Lag (% area exposed to floods) | 0.002 | 0.001 | −0.004 | −0.002 | 0.006 |
| Lag (% area exposed to droughts) | 0.003 | −0.006 *** | 0.001 ** | 0.007 | 0.99 |
| Adjusted R$^2$ | 0.99 | 0.99 | 0.97 | 0.95 | 0.97 |

*Notes:* ***, **, and * indicate statistical significance at 0.005, 0.05, and 0.1 probability levels.

**Table 7.** Regressions on flood and drought-affected areas with total per-capita GSDP.

| Explanatory Variables | Dependent Variable—ln (per-Capita GSDP) | | | | |
|---|---|---|---|---|---|
| | All States | ER | SWR | NCR | NWR |
| % area exposed to floods | 0.001 | −0.007 | 0.011 | −0.007 | 0.011 ** |
| % area exposed to droughts | −0.001 ** | 0.006** | −0.005 | 0.001 | 0.005 |
| Lag (% area exposed to floods) | 0.001 | −0.001 | −0.001 | −0.006 | 0.007 |
| Lag (% area exposed to droughts) | −0.007 ** | −0.006 * | 0.001 ** | 0.001 | −0.001 |
| Adjusted R$^2$ | 0.98 | 0.98 | 0.96 | 0.95 | 0.97 |

*Notes: ***, **, and * indicate statistical significance at 0.005, 0.05, and 0.1 probability levels.*

- A larger drought-exposure in consecutive years (current and lag drought exposed area) influence negatively on the non-agriculture sector GDP in the ER. Most of the drought-stricken areas of the ER have no significant flood risks. In addition, because of the priority received by the agriculture sector, the reduced water availability with consecutive droughts mostly affects the non-agricultural economic activities. The fact that other regions, especially the higher-income states in the SWR show no similar results may indicate better adaptation actions by the non-agriculture sectors of these regions.

- Despite the varying exposure to floods and droughts, all regions recorded significant growth in GDP per capita. The highest agricultural GDP per capita growth (4.5% annually) was in the southern and western states, followed by central, northwestern, and eastern states (2.6% annually). In the NWR, the growth of agriculture GDP per person was not significant, perhaps because of their already high level of agricultural productivity and unsustainable groundwater use. It is the main reason that substantial flood exposure in the NWR had a positive influence on non-agriculture GDP, which contributes most to the growth of total GDP. In fact, in all regions, the non-agriculture sector GDP increased at 5.5–6.7% annually.

### 3.2.2. Human Development

The changes of HDI over time across states show similar trends. Therefore, for estimating the impact on HDI, we only use a state fixed panel regression model. Results (in Table 8) show that floods generally had a negative influence on human development in all regions except in the NWR. The significant negative impact in the ER may be due to massive displacement, and its secondary impacts on health, education, and economic development. In the NWR, floods had a positive impact on the GSDP. This may be due to that floods augmented groundwater water supply, which increased access to water supply and sanitation, and also improved output in all sectors. On the other hand, lagged drought exposure had a significant negative influence on HDI. This is because droughts constrained economic growth, and hence posed an increased risk to health and nutrition. However, investments on flood mitigation may have had secondary impacts, facilitating water suuply and sanitation, and education, which are the other components of HDI.

**Table 8.** Regressions of flood and droughts affected areas with human development.

| Explanatory Variables | Dependent Variable—ln(Human Development Index) | | | | |
|---|---|---|---|---|---|
| | All States | ER | SWR | NCR | NWR |
| % area exposed to floods | −0.002 | −0.005 * | 0.017 | 0.021 | 0.015 * |
| % area exposed to droughts | −0.001 ** | −0.006 | −0.001 ** | −0.001 | −0.001 |
| Lag (% area exposed to floods) | −0.001 | −0.006 ** | 0.017 | 0.041 ** | 0.015 ** |
| Lag (% area exposed to droughts) | −0.002 *** | −0.003 * | −0.001 *** | −0.002 ** | −0.008 |
| Adjusted R$^2$ | 0.93 | 0.82 | 0.93 | 0.74 | 0.84 |

*Notes: *, **, and *** indicate statistical significance at 0.005, 0.05, and 0.1 probability levels.*

DREAs showed a negative insignificant effect on HDI. The positive coefficient of NR and CR may indicate the impact of drought adaptation investments in the region. These had a positive direct impact

on the GDP, and hence an indirect influence on health and education as well. The significant negative coefficient of lag one FLEA in the NR, CR, and NWR perhaps shows the negative effects of the wave of floods in consecutive years on all components of HDI.

## 4. Discussion

Floods are recurrent with high exposure in the ER states in India. Droughts are also frequent but expose large regions of the non-eastern states. Despite frequent exposure, both total GSDP and HDI show gradual growth, albeit at varying rates. However, floods and droughts have varying impacts on different economic sectors and regions. Despite limited impacts on the national GDP, the substantial intra-country variations give evidence on the need for boosting investments on flood and drought mitigation and adaptation measures.

Even with recurrent and large floods, there is no significant adverse effect of exposure to floods on the overall GSDP and HDI of the ER states, which are in the downstream of the Ganges River Basin. Massive investments in flood mitigation and adaptation in the ER states boost its GDP and also the HDI. Moreover, the vast spread of floods recharges groundwater, which is a critical resource for irrigation in the dry periods in the non-monsoon months.

Droughts generally impact GSDP in all states, but significantly only through the non-agriculture sector. Droughts have a significant negative impact on agriculture GSDP only in the NCR, indicating these regions' inadequate adaptation preparedness and capacity. We discuss the policy implications of the findings at the regional level.

### 4.1. Eastern Region

Flood and drought mitigation and adaptation activities are essential in the ER. Increased storage, groundwater, or surface water in the upstream, can mitigate flood impacts that are recurrent and expose large areas in the downstream of the river basins. Even with these interventions, the states in the ER can still be devastated by massive floods with climate change. Therefore, flood adaptation strategies need priority, including advanced weather advisories, flood-tolerant crop varieties, and risk-transfer mechanisms such as insurance even in the agriculture sector. Two innovative interventions are:

- The upscaling of the pilot scheme of BICSA (Bundled solutions of Index insurance with, climate information, and seeds systems ro manage Agricultural risks) conducted in the ER and Bangladesh can help here [36,37], and
- The vast spread of floods recharges aquifers naturally in the ER. However, additional recharge of floodwater in the aquifers artificially (through UTFI—underground taming of floods) can mitigate the drought risks in the non-monsoon seasons [23].

### 4.2. South and Western Region

Drought strike the SWR states the hardest, but have a short-term impact on the GSDP. The SWR states already show signs of better adaption to droughts. However, more need to be done to reduce drought impacts there.

- Promote climate safety measures such as weather index insurance, climate advisories, and agricultural inputs to mitigate drought risk [38]
- Most states can benefit from additional groundwater recharge when the rainfall is high and can benefit from surface water augmentation [23].
- Some states require long-term solutions such as diversifying of agriculture or to other non-agricultural activities that require relatively lower consumptive water needs.
- The agriculture sector can reduce irrigation demand by the introduction of advanced water management technologies such as micro-irrigation and satellite technologies for identifying water stress areas for targeted irrigation.

- Practicing climate-smart agriculture [39], with or without droughts, is crucial for these states.
- In the non-agriculture sector, return flows with utilizable quality has substantial reuse value in all sectors.

### 4.3. North and Central Region

Droughts also affect NCR agriculture significantly. However, the level of adaption there is not so strong as in the SWR. The NCR states have a lower level of SDGP and HDI higher contribution from agriculture to the total SDGP. The NCR states can reduce the impacts of droughts by

- Enhancing storage and access to groundwater use sustainably,
- Increasing water productivity than land productivity, especially in the water-scarce region [40],
- Enabling infrastructure for rapid diversification to non-agricultural economic activities can reduce the high agricultural water demand and increase water availability for for other eco-system services [41].

### 4.4. North Western Region

In the NWR, groundwater withdrawal is a key to the growth of agriculture and non-agriculture sector GSDPs. However, many areas have over-exploited aquifers, and the groundwater table is declining fast and the depletion is not sustainable.

- While artificial groundwater recharge of floodwaters [23], whenever available, is very useful, the demand management is the key to enhance resilience against droughts.
- The demand management includes reducing the area of high-water consuming crops such as rice and sugarcane, diversification to less water consuming crops including green fodder to support milk production, which is a key component in the agricultural GSDP [35]. Because the NWR states at present contribute to food security of food production deficit regions, crop diversification should be gradual to achieve sustainable groundwater use in the region.
- Efficient implementations of other soft initiatives such as delayed transplanting of rice crops, and direct benefit transfer of electricity, can induce a reduction in groundwater withdrawals and consumption leading to enhance resilience to recurrent droughts in the states [36].

### 5. Conclusions

The state-level analysis in this paper is illustrated the intra-country impacts of natural disasters (specifically floods and droughts), which global or national studies do not identify. The results show a wide variation of impacts of floods and droughts in regions and states. The findings of this study are adequate to set broad guidelines of physical and policy interventions to enhance resilience against floods and droughts in different regions and states.

Floods always receive substantial attention because of the destruction they bring to the localities that they inundate, mostly along with the riverine communities. Investments in flood mitigation mostly benefit the local communities and less distributional impacts over a large state. That is why flood exposure is even not significant in the ER states, where it hit the hardest at the local level. The large spread of floods has a significant impact on the GSDP because it helps natural recharge of groundwater, which is the source of water supply for many industries and the service sectors. At present, non-agricultural sectors (industries and services) drive the economic growth in many states, especially those in the south and west.

Whereas, droughts receive less attention but have significant spatial distribution and slow developing impacts. The large area affected by droughts has significant impacts on the agricultural and non-agriculture sectors. Adaptation to droughts, in the absence of adequate water resources, is the way forward for the growth of outputs in both the agriculture and non-agriculture sectors. However, the large spread of floods, when and where it occurs, can improve water resources and reduce drought

impacts. This is the case in the NWR states (e.g., Punjab and Haryana), where groundwater is highly over-exploited, and the economies are on unsustainable growth trajectories. The large flood spread has a positive impact there because it helps increase groundwater recharge and reduce groundwater pumping. In fact, the states such as Punjab and Haryana have some of the lowest average rainfall and potential for natural groundwater recharge.

The non-significant impacts of flood exposure do not mean that adaptation and mitigation interventions are not critical in the ER states. The investments of flood mitigation and adaptation are the sources for assisting the livelihoods of people living in these states, which have the lowest GSDP and highest poverty. However, proper targeting of interventions and investments can lead to an acceleration of growth in all economic sectors. Risk transfer schemes such as insurance can play a major role in boosting confidence for investments for growth.

Diversification of economic activities, to high-value crops in the agriculture sector, and high-value industries and services in the non-agriculture sector are the way forward for many of the SWR states. A large flood spread always helps in these states, but the frequency of large floods spreading to larger areas is also low. The value for each drop of water consumed is very high in these states. Therefore, investments for adaptation to droughts with diversified economic activities can boost total GSDP in these states.

The central and northern states, majorly located in the upper or middle part of large river basins, have no significant exposure to floods. Exposure to droughts is the main constrain for economic growth. Additional storage, water diversions, groundwater recharge, and crop diversification are the water-related interventions that can boost the economic growth in these states. Eventually, these states will also have to follow the path of western and southern states to increase growth and prosperity.

However, the results of the present study are also confounded by large intra-state variation. Many states have more than 50 million, and in some states, exceed 100 million by now. And, some states have a large area and substantial variation in hydro-geological and socio-economic conditions. Therefore, the information available for identifying spatially targeted interventions addressing mitigation and adaptation to floods and droughts is still not adequate. Only sub-national risk information at the higher spatial resolution, such as districts or blocks, ideally sub-districts or hydrological response units, would facilitate effective spatially targeted mitigation and adaptation responses. Future research should focus on profiling the risk of natural disasters at the finer spatial resolution, which will be useful for targeting better interventions that achieve faster economic growth and enhanced resilience against natural disasters.

**Author Contributions:** U.A. and G.A. conceptualized the study. N.A., S.G. and G.A. performed the remote sensing data assessment and exposure analysis. U.A. and G.A. conducted the statistical data analysis written the paper. All authors have read and agreed to the published version of the manuscript.

**Funding:** This research received no external funding.

**Acknowledgments:** This research was funded by the CGIARs (Consultative Group of International Agricultural Research Program (CRP) on Climate Change, Agriculture and Food Security (CCAFS), and CGIAR Research Program (CRP) on Water, Land and Ecosystems (WLE), which is carried out with support from the CGIAR Trust Fund and through bilateral funding agreements. For details, please visit https://ccafs.cgiar.org/donors and https://wle.cgiar.org/donors. The authors would also like to thank the Indian Council of Agricultural Research (ICAR) and Japan's Ministry of Agriculture, Forestry, and Fisheries (MAFF).

**Conflicts of Interest:** The authors declare no conflict of interest.

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
