# Peer review of "How Do Floods and Drought Impact Economic Growth and Human Development at the Sub-National Level in India?"

_climate, doi:10.3390/cli8110123_

Round 1

Reviewer 1 Report

Dear authors, your work is well designed.
To make it even better, you need to change some parts of your paper.

Abstract is poorly written. Details of research methodology are missing, details of the most important research results are missing, the most important conclusions are missing. A well-written abstract increases the visibility of your research!

Figure 1 - In the map legend the borders of the states and the borders of the basins are not defined. Only empty fields are visible.

Figure 2 - It is too small. Nothing is visible on the maps, everything is very small.

Figure 3a and 3b - On both figures there are many overlaps of markings (letters), it is difficult to read the markings. Increasing the figures along the x and y axes would probably help in the readability of the figures.

The conclusion is poorly written. Write a new conclusion that will provide the right information about the significance of the results of your research.

Author Response

Reviewer 1 comments

The abstract is poorly written. Details of research methodology are missing, details of the most important research results are missing, the most important conclusions are missing. A well-written abstract increases the visibility of your research!

We have revised the abstract to make it more readable and informative. In the methodology section, we have outlined the approach from explorative to trend analysis, and econometric analysis using state and time fixed effect panel, regression model. In the result section, we have revised the results as per the comments of another reviewer. We have also revised the conclusions, highlighting the results and policy implications.

Figure 1 - In the map legend, the borders of the states and the borders of the basins are not defined. Only empty fields are visible.

Figure 1 was redrawn with only State and River Basin boundaries

Figure 2 - It is too small. Nothing is visible on the maps, everything is very small.

The font size of the legend of Fig 2 was increased

Figure 3a and 3b - On both figures there are many overlaps of markings (letters), it is difficult to read the markings. Increasing the figures along the x and y axes would probably help in the readability of the figures.

We have decreased the size of the markers, spread out the labels to d reduce overlapping, and stretched the Y-axis to enlarge the figure.

The conclusion is poorly written. Write a new conclusion that will provide the right information about the significance of the results of your research.

We have expanded the conclusion to include the major results and their policy implications.

Reviewer 2 Report

REVIEW: How do floods and drought impact economic growth and human development at the sub-national level in India?

Abstract: The sentence seems incomplete: “Although floods expose a larger area and droughts have impacts that are more significant at the sub-national level”. I suggest replacing (.) with (,) in order to connect with the next sentence.

Introduction: The introduction is very well done. It starts from a global average analysis of the financial impacts resulting from natural disasters related to excess and scarcity of water, showing that if regionalized studies are not carried out, it becomes difficult to quantify such impacts in the long term in the development of nations. This is the objective of the research, to take this detail to specific regions of India in order to identify which are the local responses to such events and how each specific area responds more or less positively to the difficulties imposed by the climate.

Materials and Methods: Study area, data, indexes and regression models are well described. Question: On page 6, line 158, would Econometric analysis be a subtopic of the section?

Results: The results are very detailed. However, the acronyms within the bubbles in Figures 3A and 3B can confuse the reader because there is no clear association with the acronyms in the body of the text. The results of the sectoral impacts of floods and droughts are interesting, as is the positive impact of floods on the human development index due to the positive effects on water supply and agricultural production.

Discussions: The discussions are very good, but I suggest textualizing the content, removing information in topics.

Conclusions: In view of everything that was exposed in the article, in my understanding the conclusions are very succinct, and could gain more content regarding the results obtained and demonstrated.

Author Response

Reviewer 2 comments

Abstract: The sentence seems incomplete: “Although floods expose a larger area and droughts have impacts that are more significant at the sub-national level”. I suggest replacing (.) with (,) in order to connect with the next sentence.

  • This was done.

Introduction: The introduction is very well done. It starts from a global average analysis of the financial impacts resulting from natural disasters related to excess and scarcity of water, showing that if regionalized studies are not carried out, it becomes difficult to quantify such impacts in the long term in the development of nations. This is the objective of the research, to take this detail to specific regions of India in order to identify which are the local responses to such events and how each specific area responds more or less positively to the difficulties imposed by the climate.

  • We have inserted two paragraphs highlighting some of the mitigation and adaptation responses suggested and implemented at different locations.

Materials and Methods: Study area, data, indexes and regression models are well described. Question: On page 6, line 158, would Econometric analysis be a subtopic of the section?

  • Yes, it was Econometric Analysis

Results: The results are very detailed. However, the acronyms within the bubbles in Figures 3A and 3B can confuse the reader because there is no clear association with the acronyms in the body of the text. The results of the sectoral impacts of floods and droughts are interesting, as is the positive impact of floods on the human development index due to the positive effects on water supply and agricultural production.

  • We have used the acronyms to show the location of the states in the 3-dimensional graph, but highlighted results using their full names. However, we have added a note relating the acronyms with the full name.  

Discussions: The discussions are very good, but I suggest textualizing the content, removing information in topics.

  • Our objective was to keep the discussion with bullet points for highlighting the changes for different regions. We have not made any changes here except those that reflect the new analysis.

Conclusions: In view of everything that was exposed in the article, in my understanding the conclusions are very succinct, and could gain more content regarding the results obtained and demonstrated.

  • We have expanded the conclusion section by highlighting the major findings (high attention for floods, but significant impacts with droughts) and their policy implications.

Reviewer 3 Report

A study on the impacts of floods and droughts on economic growth and human development. The manuscript has two main flaws that endanger its potential publication in its current condition. See below for more details:

1) The paper lacks a literature review. There is a vast array of previous studies dealing with the impacts of floods and droughts on economic growth. The authors must provide an overview about the most significant works among these previous studies and justify how their approach outperforms what is already done. In other words, the originality, novelty and need for this manuscript must be explictly stated.

2) The methodology is not well approached. The theoretical foundations of multiple regression analysis are not well described. I strongly recommend the authors using the predicted R2 coefficient. The model may provide a good fit for the existing data, but maybe it is not as good at making predictions. Other goodness of fit measures like the Standard Error of the Regression or the Value Inflation Factor should also be studied. Not to mention the residual analysis (independence, homoscedasticity, normality and linearity), whose assessment is required to guarantee that the assumptions of multiple regression are met. By disregarding these aspects, there is no certainty about the reliability of the model and, therefore, the soundness of the results achieved.

Author Response

Reviewer 2 comments

1) The paper lacks a literature review. There is a vast array of previous studies dealing with the impacts of floods and droughts on economic growth. The authors must provide an overview about the most significant works among these previous studies and justify how their approach outperforms what is already done. In other words, the originality, novelty and need for this manuscript must be explictly stated.

  • We have inserted a review of recent literature highlighting the use of cross country penal data analysis, and only few sub-national analysis that we found. We highlight the fact our approach differs from others in using the flood and drought exposure data that shows the distributional impacts of natural disasters on natural groundwater recharge or exploitation.

2) The methodology is not well approached. The theoretical foundations of multiple regression analysis are not well described. I strongly recommend the authors using the predicted R2 coefficient. The model may provide a good fit for the existing data, but maybe it is not as good at making predictions. Other goodness of fit measures like the Standard Error of the Regression or the Value Inflation Factor should also be studied. Not to mention the residual analysis (independence, homoscedasticity, normality and linearity), whose assessment is required to guarantee that the assumptions of multiple regression are met. By disregarding these aspects, there is no certainty about the reliability of the model and, therefore, the soundness of the results achieved.

  • We have explained the approach of analysis as the state and time fixed panel data analysis. And we have explained the diagnostic statistics (endogeneity, strong instrument, model identifiability) that go along with an instrumental variable 2SLS.

Reviewer 4 Report

Dear authors i find some interesting your work, but in my opinion is necessary to improve soma aspects.

Introduction section is very messy, the quality of information is good, but is necessary to adjust some section (lines 36 -57)

I suggest to improve the quality of table 1.

The materials and methods section is very weak is necessary to describe the methodology (Which panel analysis do you use?) explain the variables in the regression and justify the choiche of the metodology through literature.

The descriptive statistics of the variable.

Lines 354 - 355, you suggest some solution, is necessary to cite the literature:

See and Cite

Spreng, C. P., Sovacool, B. K., & Spreng, D. (2016). All hands on deck: polycentric governance for climate change insurance. Climatic Change139(2), 129-140.

Broberg, M. (2020). Parametric loss and damage insurance schemes as a means to enhance climate change resilience in developing countries. Climate Policy20(6), 693-703.

Same thing for lines 368 - 380

Eze, E., Girma, A., Zenebe, A. A., & Zenebe, G. (2020). Feasible crop insurance indexes for drought risk management in Northern Ethiopia. International Journal of Disaster Risk Reduction, 101544.

Miao, R. (2020). Climate, insurance and innovation: the case of drought and innovations in drought-tolerant traits in US agriculture. European Review of Agricultural Economics.

For lines 384 -386

see and cite

Coluccia, B., Valente, D., Fusco, G., De Leo, F., & Porrini, D. (2020). Assessing agricultural eco-efficiency in Italian Regions. Ecological Indicators116, 106483.

In the conclusion section is necessary to evidence the implication of your policy suggestions.

Mariani, M., Fletcher, M. S., Haberle, S., Chin, H., Zawadzki, A., & Jacobsen, G. (2019). Climate change reduces resilience to fire in subalpine rainforests. Global change biology25(6), 2030-2042.

Jakob, M., Lamb, W. F., Steckel, J. C., Flachsland, C., & Edenhofer, O. (2020). Understanding different perspectives on economic growth and climate policy. Wiley Interdisciplinary Reviews: Climate Change, e677.

I reccomended a major revision

Author Response

Review 4 comments

Introduction section is very messy, the quality of information is good, but is necessary to adjust some section (lines 36 -57)

  • We have revised the introduction section to address reviewers concerns.

Sugest to improve the quality of table 1.

  • We have not revised Table 1, as we wanted to show the major differences across states.

The materials and methods section is very weak is necessary to describe the methodology (Which panel analysis do you use?) explain the variables in the regression and justify the choiche of the metodology through literature.

  • The methodology section was revised to show that we first use an explorative analysis and then to assess basin trends of dependent variables; And finally, the econometric model of 2SLS.

The descriptive statistics of the variable.

Lines 354 - 355, you suggest some solution, is necessary to cite the literature:

See and Cite (used the following highlighted literature)

Spreng, C. P., Sovacool, B. K., & Spreng, D. (2016). All hands on deck: polycentric governance for climate change insurance. Climatic Change139(2), 129-140.

Broberg, M. (2020). Parametric loss and damage insurance schemes as a means to enhance climate change resilience in developing countries. Climate Policy20(6), 693-703.

Same thing for lines 368 - 380

Eze, E., Girma, A., Zenebe, A. A., & Zenebe, G. (2020). Feasible crop insurance indexes for drought risk management in Northern Ethiopia. International Journal of Disaster Risk Reduction, 101544.

Miao, R. (2020). Climate, insurance and innovation: the case of drought and innovations in drought-tolerant traits in US agriculture. European Review of Agricultural Economics.

For lines 384 -386

see and cite

Coluccia, B., Valente, D., Fusco, G., De Leo, F., & Porrini, D. (2020). Assessing agricultural eco-efficiency in Italian Regions. Ecological Indicators116, 106483.

In the conclusion section is necessary to evidence the implication of your policy suggestions.

Mariani, M., Fletcher, M. S., Haberle, S., Chin, H., Zawadzki, A., & Jacobsen, G. (2019). Climate change reduces resilience to fire in subalpine rainforests. Global change biology25(6), 2030-2042.

Jakob, M., Lamb, W. F., Steckel, J. C., Flachsland, C., & Edenhofer, O. (2020). Understanding different perspectives on economic growth and climate policy. Wiley Interdisciplinary Reviews: Climate Change, e677.

Round 2

Reviewer 3 Report

The authors are commended for their efforts to improve the manuscript. The document does have substantially improved in what concerns the inclusion of literature review and the elaboration of the methods used. 

Reviewer 4 Report

Well done, in my opinion is suitable for a publication!